

# 3D-Var versus Optimal Interpolation for Aerosol Assimilation: a Case Study over the Contiguous United States

Youhua Tang[1,2] (youhua.tang@noaa.gov), Mariusz Pagowski[4,5], Tianfeng Chai[1,2], Li Pan[1,2], Pius Lee[1], Barry Baker[1,2], Rajesh Kumar[6], Luca Delle Monache[6], Daniel Tong[1,2,3], and Hyun-Cheol Kim[1,2]

1. NOAA Air Resources Laboratory, College Park, MD.
2. Cooperative Institute for Climate and Satellites, University of Maryland at College Park, MD.
3. Center for Spatial Information Science and Systems, George Mason University, Fairfax, VA.
4. NOAA Earth System Research Laboratory, Boulder, CO.
5. Cooperative Institute for Research in Environmental Sciences, University of Colorado, Boulder, CO.
6. National Center for Atmospheric Research, Boulder, CO.

## Abstract

This study applies the Gridpoint Statistical Interpolation (GSI) 3D-Var assimilation tool originally developed by the National Centers for Environmental Prediction (NCEP), to improve surface $PM_{2.5}$ predictions over the contiguous United States (CONUS) by assimilating aerosol optical depth (AOD) and surface $PM_{2.5}$ in version 5.1 of the Community Multi-scale Air Quality (CMAQ) modeling system. GSI results are compared with those obtained using the optimal interpolation (OI) method (Tang et al., 2015) for July, 2011 over CONUS. Both GSI and OI assimilate surface $PM_{2.5}$ observations at 00, 06, 12, and 18UTC, and MODIS AOD at 18 UTC. In the GSI experiments, assimilation of surface $PM_{2.5}$ leads to stronger increments in surface $PM_{2.5}$ compared to the MODIS AOD assimilation. In contrast, we find a stronger impact of MODIS AOD on surface aerosols at 18 UTC compared to the surface $PM_{2.5}$ OI assimilation. The increments resulting from the OI assimilation are spread in 11×11 horizontal grid cells (12km horizontal resolution) while the spatial distribution of GSI increments is controlled by its background error covariances, and the horizontal/vertical length scales. The assimilations of observations using both GSI and OI generally help reduce the prediction biases, and improve correlation between model predictions and observations. GSI produces smoother result and yields overall better correlation coefficient and root mean squared error (RMSE). In this study, OI uses the relatively big model uncertainties, which helps yield better mean biases, but sometimes causes the RMSE increase due to its localized correction. We also examine and discuss the sensitivity of the assimilation experiments results to the AOD forward operators.

## 1. Introduction

The existing U.S. National Air Quality Forecasting Capability (NAQFC) run by the National Oceanic and Atmospheric Administration (NOAA)/National Centers for Environmental




Prediction (NCEP)  provides daily 48-hour  ozone and $PM_{2.5}$ (particle matters with diameter <
2.5 μm) forecasts using the Community Multi-scale Air Quality (CMAQ) modeling system with
12km horizontal grid resolution. Many contributions toward improving the NAQFC can be
found in literatures, including updated emission, meteorology, chemical mechanism and lateral
boundary conditions (Pan et al., 2014; Tang et al., 2008; Lee et al., 2016). However, biases still
contaminate current predictions.
Chemical data assimilation techniques have been developed to improve initial conditions of
chemical transport models (CTM) and yield better prediction ((Elbern et al., 1997, 2000, 2007;
Elbern and Schmidt, 1999, 2001; Bocquet et al., 2015) by blending the information from a model
estimate (refer to as prior or background) and from observations in certain methods. One method
is direct blending, e.g. using observed chemical mass concentrations to correct the modeled mass
concentrations, which is relatively straightforward, as they are directly comparable. As most
monitoring data are located near surface, that method was usually applied to near-surface field.
Another method is indirect guessing, e.g. comparing satellite retrieved AOD with modeled AOD
to estimate the biases of modeled column mass concentrations and make the corresponding
adjustment. Tang et al. (2015) used both surface $PM_{2.5}$ and MODIS AOD to adjust the initial
condition of a CTM with the optimal interpolation (OI) method, and successfully reduced the
model biases. The OI correction was applied to 7×7 or 11×11 grid cells horizontally, and from
the ground to the boundary layer top vertically for surface $PM_{2.5}$ assimilation. Adhikary et al.
(2008) and Chai et al. (2017) also used the similar local OI correction to reduce model biases. A
more complex method is using the Gridpoint Statistical Interpolation (GSI) (Wu et al., 2002;
Purser et al., 2003a, b), a 3D-VAR method, developed by NOAA/NCEP.  It has been applied to
assimilate the Goddard Chemistry Aerosol Radiation and Transport (GOCART) aerosols in
Weather Research and Forecasting model coupled with Chemistry (WRF-CHEM) with its 3D-
var method (Pagowski et al., 2014; Liu et al., 2011). In next sections, we describe the method
that extends the GSI to assimilate CMAQ aerosols comparing to the OI correction. A comparison
for their one-month performances will be discussed.

## 2. Methodology and Settings

The baseline model setting used in this study is similar to Tang et al. (2015), except that the
CMAQ model changed from version 5.0.2 to version 5.1 for more refined chemical mechanisms
and physical schemes. The meteorology is provided by the WRF-ARW (version 3.4.1) driven by
the NCEP FNL (Final Global Analysis) 1° × 1° analysis field and was reinitialized every 24
hour.  Both the meteorological and air quality models have 12-km horizontal resolution over the
contiguous United States, with 42 sigma layers vertically, and the domain tops at 50 hPa. The
detailed setting of the CMAQ model can be found in Tang et al. (2015). In order to compare the
two assimilation methods, we assimilate same surface $PM_{2.5}$ data (USEPA Air Quality System



(AQS) data) 4 times every day (00Z, 06Z, 12Z and 18Z). The 18Z assimilation uses additional
MODIS AOD data from Terra and Aqua satellites.

## 2.1 Settings for Data Assimilations

In this study, we are comparing two assimilation methods: the optimal interpolation and GSI's
3D-Var. The optimal interpolation is carried in the similar way as described in the OI4 case of
Tang et al. (2015) for assimilating surface $PM_{2.5}$ and aerosol optical depth retrieved from
Aqua/Terra MODIS sensors.

$$X^a = X^b + BH^T(HBH^T+O)^{-1}(Y-HX^b) \tag{1}$$

where $X^a$ and $X^b$ are the analyzed and background (prior modeled) concentrations or aerosol
optical depth (AOD) data, $B$ and $O$ are the background and observation error covariance
matrices, $H$ is the observational operator and $H^T$ is its matrix transpose, and $Y$ is the observation
vector. The relative uncertainty setting is also same as Tang et al. (2015), in which the
background relative uncertainties have horizontal and diurnal variations. The surface $PM_{2.5}$ OI is
applied from surface to the height of the planetary boundary layer (PBL), and the MODIS AOD
is used to adjust above-PBL aerosol after deducting the adjusted below-PBL AOD.
The GSI's 3D-var uses a similar approach (Pagowski et al., 2014; Liu et al., 2011) for its cost
function

$$J = \frac{1}{2}(x_a-x_b)^T B^{-1}(x_a-x_b) + \frac{1}{2}(Hx_a-O_o)^T O^{-1}(Hx_a-O_o) + Jc \tag{2}$$

Where $x_a$ and $x_b$ are the analyzed and background (a prior modeled) concentrations or AOD data,
$B$ and $O$ are the background and observation error covariance matrices, $H$ is the observational
operator, $O_o$ is the observations and $J_c$ is the constraint terms. Both GSI's 3D-Var and OI use
spatially varied background bias and observation to make the adjustment. However, the OI
adjustment is made in each 11×11 grid horizontally and its effect expands up to the PBL height.
The GSI's cost function reduction is performed for the whole domain, and its effect of the
adjustment can be expanded in much greater horizontal and vertical scales defined by its
horizontal and vertical length scales, respectively.
In this study, we carry out the similar uncertain setting for the OI as Tang et al. (2015), in which
the background's relative uncertainties have horizontal and diurnal variations (Figure 2 of Tang
et al., 2015) for PM2.5 assimilation. For OI's AOD assimilation, we use the fixed relative
uncertainty of 0.8 for modeled AOD. The relatively uncertainties for MODIS AOD and surface
$PM_{2.5}$ observation are same: 0.1. GSI's setting is similar to Pagowski et al. (2014), in which the
background error and length scales have vertical variance. Figure 1 shows that the vertical



profiles of background errors and length scales for PM$_{2.5}$ (for PM$_{2.5}$ assimilation) and
accumulation-mode sulfate (ASO4J, for AOD assimilation), which can be calculated using GSI's
NMC (National Meteorological Center) method (Parrish and Derber, 1992). ASO4J is one of
CMAQ aerosol species used in AOD data assimilation. Other aerosol species have proportional
model or background errors. The uncertainties of modeled aerosols are high in the lower layers
and decrease with altitudes. The model's horizontal length scale indicates the extent of the
assimilation's horizontal expansion. In Figure 1, the horizontal length scale for modeled PM$_{2.5}$
increases significantly around the altitude of 12km, where the tropopause is usually located. The
model's vertical length scale indicates the extent of the assimilation's vertical expansion, which
is related to the strength of vertical advection, diffusion, and convection. Below PBL, the vertical
length scale is usually stronger than that in the upper layers. We use the same constant
observation error of 0.1 in GSI for surface PM$_{2.5}$ and MODIS AOD. In both settings of GSI and
OI, their background errors are far greater than the observation errors in lower altitudes, which
push the adjusted values toward the observed surface PM$_{2.5}$. For AOD assimilations, GSI's main
increment is also in low altitudes due to its background error profile (Figure 1), while OI's AOD
assimilation applies one adjusting ratio to whole column for each grid cell.
**2.2 Calculations of Aerosol Optical Depths**
To assimilate AOD in the CMAQ model, we need convert CMAQ aerosol chemical
compositions. CMAQ includes two methods for calculating AOD: the Mie method and the
reconstruction method (Binkowski and Roselle, 2003). The Mie method calculates the aerosol
optical extinction coefficient (AOE) from modeled aerosol physical characteristics, including the
index of refraction, volume concentration and aerosol size distributions. It does not require
aerosol chemical composition information and can handle the aerosols' internal mixture, in
which each aerosol particle is composed of a solid core and coating layers of various
compositions due to the aerosol uptake. In theory, the Mie method should yield accurate AOE if
all the model's aerosol physical properties are correct. However, that condition was hard to reach
in many circumstances. Also, aerosol mass concentrations are often available in both models and
observations, and a convenient method is needed to directly convert aerosol mass concentrations
to AOE. So, CMAQ provides another empirical approach, the reconstruction method (RM),
which uses the mass concentrations of aerosol chemical compositions to calculate the total AOE
with a look-up table. The RM assumes that all aerosols are externally mixed. It calculates each
composition's AOE, and sums them up to get the total AOE (Binkowski and Roselle, 2003) for
the wavelength of 550nm.
$AOE \left[\frac{1}{km}\right] = 0.003 * f(RH) * \{(NH_4)_2SO_4 + NH_4NO_3\} + 0.004 * \{oragnic\ mass\} + 0.01 *$
$\{light\ absorbing\ carbon\} + 0.001 * \{fine\ soil\} + 0.0006 * \{coarse\ mass\} + 0.00137 *$
$f_s(RH) * \{sea\ salt\}$   (3)





In CMAQ's RM, only the AOE from ammonium, sulfate, nitrate, and sea salt have the
dependence on relative humidity (RH). $f(RH)$ and $f_s(RH)$ are two RH-dependent look-up tables
of aerosol hydroscopic growth for sulfate/nitrate/ammonium and sea salt, respectively (the
original RM (Binkowski and Roselle, 2003) in earlier versions (<5.1) of CMAQ does not include
sea salt's RH dependence for AOE calculation). All other aerosols are converted from their mass
concentrations to the corresponding AOEs with simple fixed constants. The conversion method
does not need aerosol size distribution, but just the mass concentrations of aerosol compositions
and ambient RH. Due to its simplicity and convenience, the RM method is widely used not only
for CMAQ aerosols, but also for converting observed aerosol mass concentrations to AOE
(Malm et al., 1994; Roy et al., 2007). Tang et al. (2015) also use the RM calculated AOD for OI
assimilation, which is carried out here for this OI assimilation. In contrast, the Mie method based
on aerosol physical characteristics is relatively hard to be used in data assimilations, as the data
assimilations target the mass concentrations of aerosol compositions, not the aerosol physical
characteristics directly. Although we can use adjusted aerosol mass concentrations to calculate
the corresponding change on aerosol physical characteristics with CMAQ routines, and then link
it to AOD adjustment, the additional conversion makes these calculations difficult.
In theory, we should use the same forward method, such as RM, to calculate AOD used in GSI.
However, the CMAQ's current RM is relatively simple, only for single wavelength, 500nm,
without considering aerosol distribution. The current RM can not be directly used to calculate
multi-wavelength AOD, though this study only uses the 550nm AOD. The existing GSI has its
own tool, the Community Radiative Transfer Model (CRTM, Han et al., 2006; Liu and Weng,
2006), to handle its AOD calculation. CRTM provides GSI not only the forward AOD, but also
the calculations of tangent-linear, adjoint and K-matrix for multiple wavelengths. As the GSI is
tightly coupled with CRTM for its AOD related operations, we need to go through the CRTM for
utilizing GSI's existing AOD assimilation capability. However, the current version of CRTM
does not support CMAQ's aerosol species, and only handle GOCART (Goddard Chemistry
Aerosol Radiation and Transport, Chin et al., 2000, 2002) aerosol species. GOCART's aerosol
species are similar to the classification of aerosol used in CMAQ RM method, including sulfate,
dust, sea salt, black carbon, and organic carbon, which also assumes that the aerosols are
externally mixed. It is possible for us to represent CMAQ aerosols in GOCART aerosol
categories for calculating their optical properties. Table 1 shows how the CMAQ 5.1 Aero6
species (Sonntag et al., 2014) are mapped to CRTM's GOCART aerosol.  CMAQ aerosols have
3 size modes: Aitken (i-mode), accumulation (j-mode) and coarse mode (k-mode) representing
the super-fine nucleation particles, aged coagulated particles, and coarse particles, respectively
(Binkowski and Roselle, 2003). Each size mode has its own lognormal size distribution (Whitby
and McMurry, 1997). We applied the CMAQ averaged aerosol size for each mode (aitken,
accumulation or coarse) to the CRTM AOD calculation.  Unlike the CMAQ RM method in
which only AOEs from sulfate, nitrate, ammonium and sea salt have RH dependence, all the
calculations for GOCART AOEs except hydrophobic BC/OC depend on ambient RH,
wavelength and aerosol sizes, which differs from RM's one-size-fit-all method.





Figure 2 shows one example of the AOD calculations from the 3 methods mentioned above
compared to Terra/Aqua MODIS AOD data. The MODIS AODs in Figure 2 are the daily data,
and the overpass times of Terra and Aqua satellites on 07/01/2011 ranges from 15 to 22 UTC
over the CONUS. Both OI and GSI use the AOD assimilation window of +/- 2 hours. During this
event, some wildfire occurred in Southern Canada-Wisconsin, North and South Carolinas, which
caused the relatively high AOD values. These high AODs are also confirmed by the MODIS
retrievals (Figure 2d). Figure 2 shows that the different AOD calculations from the same CMAQ
aerosol mass loadings yield the similar spatial distribution pattern with noticeable quantitative
differences, and in general we see RM AOD > Mie AOD > CRTM AOD. In most regions,
especially over Western USA, e.g. Nevada, MODIS AOD is higher than the three CMAQ AODs.
The Mie-method AOD and CRTM AOD are generally lower than MODIS AOD. Only RM
method shows some sporadic overestimated AOD over Southern Canada-Wisconsin and
Carolinas. These differences will lead to the corresponding differences in data assimilation
adjustments.  Roy et al. (2007) compared the CMAQ AOEs with surface AOE observations from
the nephelometer instrument over 25 Interagency Monitoring of Protected Visual Environment
(IMPROVE) sites (most of them were located in National Parks), and found that the CMAQ RM
method yielded too high surface AOE, but agreed well with AEORNET (Aerosol Robotic
Network) and MODIS AOD in summer 2001. In Roy et al. (2007), CMAQ used static lateral
boundary condition, which may miss some elevated aerosol loadings (Lu et al., 2016). It is very
likely that their CMAQ overpredicted near-surface AOD loading and underpredicted elevated
AOD loading, but got total column AOD in right magnitude. In another word, the converting
factors of RM method used in equation (3) may be too high if their CMAQ predicted correct
aerosol mass concentration over those National Park sites.

### 2.3 PM$_{2.5}$ Calculations

Besides the MODIS AOD, The data assimilations of OI and GSI also use surface PM$_{2.5}$
observations to make adjustments. The OI method for surface PM$_{2.5}$ assimilation was described
in Tang et al. (2015). Pagowski et al. (2014) described the GSI method for surface PM$_{2.5}$
assimilation used in WRF-CHEM in which aerosol size bins are fixed. CMAQ's PM$_{2.5}$
calculation is slightly different as it is not defined by fixed size bins, but three size modes:
Aitken, accumulations and coarse modes (i, j, k modes) (Appel et al., 2010).

$$PM_{2.5} = PM25AT * (SO4_i + NO3_i + NH4_i + Na_i + Cl_i + EC_i + POC_i + OTHER_i) +$$
$$PM25AC * (SO4_j + NO3_j + NH4_j + Na_j + Cl_j + EC_j + EL_j + POC_j + SOA_j + OTHER_j) +$$
$$PM25CO * (SO4_k + NO3_k + NH4_k + Na_k + Cl_k + SOIL_k + OTHER_k) \quad (4)$$

where PM25AT, PM25AC and PM25CO are the mass scaling factors for the three modes,
indicating the mass ratios of each mode falling in PM$_{2.5}$ size range (Jiang et al., 2006), and their
value ranges should be between 0 and 1. POC is the primary organic carbon, and SOA represents
the 23 secondary organic aerosols (table 1). CMAQ 5.1 also includes 8 additional mineral
elements: Fe, Al, Si, Ti, Ca, Mg, K and Mg (Table 1) in its accumulation mode, represented by





*EL_j*. The species *OTHER* represent all other aerosols (Table 1) in each mode. The equation 4 describes the forward operator for calculating $PM_{2.5}$, which is feed into GSI as background $PM_{2.5}$ concentration. After the GSI estimates the total $PM_{2.5}$ increment, the increment is proportionally allocated to each aerosol size mode with the corresponding mass scaling factor, and then to each aerosol species. Thus, the final adjustment that OI and GSI give back to CMAQ is the mass concentration increment for each aerosol species, and it does not change their size distributions, which is also the effect of AOD assimilation.

## 3. Results and Discussions

As the GSI and OI methods assimilate both surface $PM_{2.5}$ and AOD, the impacts of these adjustments can be compared. Figure 3 shows the CMAQ raw predictions (referred to as base-case run) of surface $PM_{2.5}$ compared to the measurements. In most regions west of 100ºW, this a priori case underestimated the surface $PM_{2.5}$. Surface measurement also shows some sporadic high $PM_{2.5}$ values in Eastern USA, missed by the base-case run. The goal of the data assimilations is to reduce these errors.

### 3.1 The Impact of Data Assimilation on Aerosol Mass Concentration

Since both $PM_{2.5}$ and aerosol AOD represent the concentrations of 54 aerosol species (Table 1), to understand the data assimilation method performances, we choose the accumulation-mode or J-mode sulfate (ASO4J), to illustrate the impact of data assimilation. Other aerosol species have proportional adjustments in the GSI and OI assimilations. Figure 4 (a, b) shows that the GSI assimilation with surface $PM_{2.5}$ yields more significant changes than the corresponding OI assimilation. The OI assimilation of $PM_{2.5}$ leads to more localized increments compared to GSI since OI increment is spread over $11\times11$ grid cells (i.e., an area of 17424 $km^2$), while GSI increment spread depends on the horizontal length scales. Their adjustments also differ over certain areas. For instance, the base case underestimates $PM_{2.5}$ in Chicago, but it overestimates $PM_{2.5}$ in southeastern Wisconsin (Figure 3). The OI adjustment has a local flavor: increase ASO4J in Chicago, and decrease it over southeastern Wisconsin (Figure 4b). The GSI's result differs significantly due to its length scales and cost-function reduction over the whole domain. It yields overall $PM_{2.5}$ reduction in Chicago surrounding areas with smaller correction. Similar differences are also noted in other areas and other periods.

The impacts of AOD assimilations via GSI and OI also differ significantly (Figure 4c, 4d). The model relative uncertainty used in the OI assimilation is relatively high, which caused the strong adjustment in ASO4J (Figure 4d) due to OI's AOD assimilation over Nevada-Southern California, Illinois and Michigan. The OI assimilation of AOD increases ASO4J over most of the model domain except in certain areas, such as Northeastern Minnesota-Southern Canada, and the eastern part of the border of North Carolina and South Carolina where the OI assimilation leads to a decrease in ASO4J (Figure 4d). The GSI assimilation of AOD tends to be more moderate and smoother. Its increment is almost one order of magnitude smaller than the variation of OI-



AOD assimilation (Figure 4c). Figure 4c shows that the majority of increments of GSI-AOD
assimilation occur in Eastern California-Nevada, Missouri-Illinois (St. Louis surrounding areas)
and Wisconsin.
Figures 4e and 4f show the combined effect of assimilation of both surface $PM_{2.5}$ and AOD on
surface ASO4J from GSI and OI methods, respectively. For GSI, the impact of assimilating
surface $PM_{2.5}$ is greater than that of assimilating MODIS AOD. The OI assimilations show the
opposite effect with AOD assimilation having a bigger impact than its surface $PM_{2.5}$
assimilation.
It should be noted that the difference between GSI and OI increments is not only on their
horizontal distribution, but also on their vertical distribution. GSI has a height-dependent
background error profile (Figure 1) while OI applies the uniform adjustment ratio to either whole
column for AOD assimilation or below PBL for surface $PM_{2.5}$ assimilation. Figure 5 shows the
model predicted PBL height at this time. In most portions of CONUS domain except the Rocky
mountain region, the PBL height is less than 3 km. It means that the impact of OI-$PM_{2.5}$
assimilation should be below 3 km for most regions. Considering the locations of available
surface $PM_{2.5}$ monitoring stations and the regional distribution of background $PM_{2.5}$, the vertical
extension of the increments of OI-$PM_{2.5}$ should be more limited. Figure 6 shows the ASO4J
increments similar to Figure 4 but for the model's 18[th] layer, or roughly 2 km above ground.
Figure 6b shows that the OI-$PM_{2.5}$ adjustment at the 2 km layer only appears in sporadic
locations, such as Raleigh/Durham (central North Carolina), Atlanta and Denver downwind
areas. Compared to Figure 4b and 5, we can find that in the most CONUS regions, the OI-
$PM_{2.5}$'s ASO4J increments do not show up in the 2km layer because either their PBLs are lower
than 2km, or there is no strong surface adjustment due to the locations of monitoring stations. On
the contrary, GSI-$PM_{2.5}$ assimilation shows the horizontal distribution of ASO4J increment at
2km similar to its surface increment (Figure 6a, 4a), but with a much smaller magnitude than the
increment of OI-$PM_{2.5}$ assimilation at the 2 km layer. The GSI-AOD assimilation also yields a
similar ASO4J adjustment pattern with smaller increments than OI-AOD (Figure 6c, 4c). The
strongest adjustment at the 2 km layer comes from the OI-AOD assimilation, which uses the
same adjustment ratio for the whole vertical column. Its impact on 2 km ASO4J is more than one
order of magnitude stronger than the corresponding GSI assimilation due to its relatively high
uncertainty setting. In term of combined effect at the 2km, for GSI, the AOD assimilation is
almost equally important as the surface $PM_{2.5}$ assimilation, depending on regions (Figure 6e).
For OI assimilation, the AOD's impact is dominated at the 2km layer as the surface $PM_{2.5}$'s
effect only appear sporadically (Figure 6f).
**3.2 The Impact of Data Assimilation on AODs**
The above discussion is about the data assimilations' impacts on one aerosol mass concentration.
It is also still needed to assess the impacts on column AODs, as the AOD is used here to make
assimilations. Fortunately both CRTM and RM are composition-based methods for externally





mixed aerosols, and we can easily calculate the AOD changes due to the changes of aerosol mass
concentrations. Figure 7 shows the CRTM AOD changes due to GSI assimilation (left panel),
and RM AOD changes due to OI assimilation (right panel). Their spatial distribution patterns are
very similar to the corresponding surface ASO4J increments in Figure 4 as most high aerosol
loadings are near surface. However, the AOD increment's value range of GSI-PM$_{2.5}$ assimilation
is more than one order of magnitude lower than that of OI-PM$_{2.5}$. One reason is that the CRTM
method yields 2 or 3 times lower AOD than the RM method with same aerosol loading (Figure
2b, 2c). Another reason, or the major reason, is that GSI-PM$_{2.5}$ assimilation has much lower
increment on total aerosol mass loading, reflected by the magnitude difference between Figure
6a and 6b, as the GSI assimilation uses a steepen background error profile (Figure 1),  makes its
major adjustment near surface and yields smaller overall adjustment for total column aerosol
mass loading. On the contrary, the OI-PM$_{2.5}$ assimilation applies the same adjustment ratio to the
aerosol masses below PBL, and the adjustment could be much stronger than that of GSI-PM$_{2.5}$
assimilation for the elevated layers blow PBL. We can see the similar patterns and differences
due to their AOD assimilations (Figure 7c, 7d). The AOD increments due to GSI-AOD and OI-
AOD have the similar horizontal distribution to their corresponding ASO4J mass increments
(Figure 4c, 6c and 7c; Figure 4d, 6d, and 7d), but the OI-AOD has much stronger adjustments on
total AOD and elevated aerosol masses. Generally, the GSI-AOD tends to increase AOD or
aerosol column loading in Northern Central USA and over Nevada (Figure 7c). The OI-AOD
assimilation has the similar AOD increment over these two regions, but decrease AOD or aerosol
column loading over Texas, Northern Florida, South Central Canada, and the border of North
Carolina and South Carolina (Figure 7d). It should be noted that the OI-PM$_{2.5}$ assimilation yields
the AOD enhancements in Southern and Eastern USA, though just sporadically (Figure 7b). So
there is a conflict as the OI-PM$_{2.5}$ and OI-AOD assimilations pointed to opposite adjusting
directions over some areas, implying that the RM AOD could yield some overpredictions. Under
this situation, the combined OI assimilations will increase below-PBL aerosol masses according
to the adjustment of OI-PM$_{2.5}$, but reduce the above-PBL aerosol mass to fit the overall AOD
reduction and get compromised results over these two regions (Figure 7f). This conflict-
resolving process will change the vertical distribution of CMAQ aerosols. Figure 7 shows that
the overall AOD increments are mainly due to PM$_{2.5}$ assimilation in GSI, and AOD assimilation
in OI. The OI adjustments on AOD are much stronger than that of GSI, which is mainly due to
their adjustments at the elevated layers.

**3.3 The Overall Assimilation Impacts over Longer Periods**

We continue the CMAQ runs after the 18UTC assimilations. After 1 hour, or at 19UTC, we
compared their surface PM$_{2.5}$ prediction with the corresponding measurement again to see their
impacts (Figure 8). At this time, the effects of data assimilations have been transported to
downstream areas. Without the data assimilation, the base case (Figure 8a) systematically
underestimated the PM$_{2.5}$ in western USA, which is consistent with that shown in Figure 3. Both
assimilations correct some of the biases, which is more obvious in areas west of 90ºW.  For



instance, both assimilations correct the PM$_{2.5}$ underprediction in Iowa. In some locations, OI gets
better result, such as in Southern Nevada-Southern Utah, and Kanas where only one surface
observation is available. In other locations, the GSI achieves better results, such as Southeastern
Wisconsin.Sometimes the data assimilations could overcorrect and yield worse results than the
base run, such as the OI's overestimation over the Lake Michigan and GSI's overprediction over
central Illinois (Figure 8b, 8c). The overcorrection issue is more evident in the OI run as the
adjustment of the OI assimilation is stronger than that of GSI, due to OI's stronger setting for
model uncertainties. This strong setting is actually helpful sometimes, for instance, the OI
assimilation is strong enough to correct the underpredicting bias over southeastern North
Carolina, where GSI's moderate correction only helps reduce that bias. The most evident side
effect of the OI's overcorrection is the increase of root mean squared error (RMSE).
In this study, we employ 4-cycle per day data assimilations, the MODIS AOD assimilation is
only applied at the cycle of 18 UTC for both the GSI and OI. We continue these runs for the
whole July 2011. Figure 9 shows the time-series plots of these CMAQ predictions for surface
PM$_{2.5}$, their correlation coefficient (R) and RMSE. Before July 14 or Julian day 195, the CMAQ
base prediction systemically underpredicted the PM2.5. After that date, the base model tends to
underpredict during daytime, but slightly overpredict during nighttime, except for the last 3 days
of July 2011 while the overall undeprediction appeared again. Both GSI and OI help reduce these
undeprediction biases. The OI's correction is stronger due to its stronger setting. It became more
evident during the PM$_{2.5}$ peak period caused by the firework emission at night of July 4$^{th}$ (U.S.
Independence Day), which is about early morning of July 5$^{th}$ (Julian day 186) in UTC time
(Figure 9a). The OI assimilation caught the firework caused PM2.5 spike though its peak time
was later than that of the observation. The GSI assimilation showed the moderate correction
which was not strong enough to full match with observation. The OI run shows some
overprediction during the nighttime, especially in later July. Figure 9b clearly shows the effect of
4 assimilations per day, represented by the 4-time-per-day enhancement of GSI and OI's
correlation coefficients. For the most of this month, we can generally see GSI>OI>CMAQ base
for R. The difference of R between GSI and OI is much smaller than that between the
assimilations runs and the CMAQ base case. GSI's R is consistent better than the CMAQ base
while the OI's R is not always better, e.g. during the Julian days of 191 and 207, though we can
still find several periods when OI yielded the highest R. In term of RMSE, the GSI's RMSE is
always lower than that of the base run, and the OI run shows some RMSE spikes (Figure 9c) due
to its localized correction and strong settings.
Table 2 shows the corresponding statistics for the whole domain and the certain regions. The
CMAQ base case systematically undepredicted surface PM$_{2.5}$ with mean bias of -2.25 µg/m$^3$
over CONUS during this period. With data assimilations, the OI and GSI runs get mean biases of
0.77 and -0.73 µg/m$^3$, respectively. Besides that effect, their correlation coefficient, R, is also
improved: the base case's R is 0.3, and Rs of OI and GSI runs are 0.38 and 0.44 over CONUS,
respectively. Their indexes of Agreement (IOA) have the corresponding improvements, too.



Among the regions, the data assimilation yield most significantly improvements over the Pacific Coast and Southeastern USA, where the CMAQ base case has relatively poor correlation coefficients. In all of these regions, the GSI yields overall best correlation coefficient and RMSE, while the OI run has the smallest mean biases except for South Central region where the GSI's mean bias is the lowest. OI's localized correction and strong settings help get better mean biases, but cause the overcorrection issue and increase RMSE. Over Northeastern USA where relatively dense surface observations are available, the OI's RMSE increase is relatively small. Although both assimilations generally improve the $PM_{2.5}$ prediction, their performance can be highly varied depending on regions. For instance, over Rocky Mountain States where both surface observation and MODIS AOD are limitedly available with complex terrains, the GSI and OI assimilations slightly improve the R and IOA, though they evidently improve the mean bias of surface $PM_{2.5}$.

All these runs have daytime underprediction bias issue from 12 to 00UTC. The 4-cycle-per-day data assimilations help reduce that bias, but the bias trend is still there. It implies that the data assimilation for CMAQ's initial condition has certain limitation, and may not be able to solve the prediction bias by it alone. In addition to uncertainties in initial conditions, forecasts errors also depend on errors in model's meteorology, chemistry and emissions, and we need other adjustments to correct these biases.

## 4. Conclusions

In this study, we expanded the GSI assimilation to CMAQ 5.1's Aero6 aerosol species. The 4-cycle-per-day aerosol data assimilation for surface $PM_{2.5}$ and AOD were carried out with GSI and OI (Tang et al., 2015) methods over the CONUS. The results were compared against surface $PM_{2.5}$ observation, and shows that both assimilations generally improved the aerosol predictions. The increments resulting from the OI assimilation are spread in 11×11 horizontal grid cells while the increment spread in GSI (a 3D-Var assimilation technique) is controlled by its background error variances, horizontal and vertical length scales. GSI's cost function reduction is performed for the whole domain. The differences in formulation of GSI and OI led to their different patterns of adjusting the initial conditions, and GSI yielded smoother and horizontally broader adjustment, but much weaker vertical increment. OI uses a simple one-ratio-fit-all method for its vertical adjustment up to PBL height for $PM_{2.5}$ assimilation, or whole column for AOD assimilation. OI can use strong setting to achieve better mean bias, but has side effect of RMSE increase due to its localized correction. Overall, GSI's adjustment yield better results even with its moderate setting of data assimilation parameters, showed by its better statistics. One important reason is that GSI's whole-domain cost function reduction, which helps constrain its RMSE, and longer horizontal length scale, especially for $PM_{2.5}$ assimilation (Figure 1), which helps expand its adjusting effect to relatively broad areas. Both assimilations highly depend on





the available observations. Compared to GSI's massive code, the OI code is much smaller and
portable, which may be good choice of some light-duty usages.
AOD assimilations have more issues than the surface $PM_{2.5}$ assimilation, which is not only about
the methods for converting aerosol mass concentrations to AOD that relies on the model for
ambient RH, aerosol sizes and speciation, but also depends on a prior model for the aerosol
vertical distribution. The CRTM AOD is about 2-3 times lower than RM AOD with the same
aerosol mass loadings (Figure 2). From the existing evidence, the converting factors used in RM
AOD may be too high (Roy et al., 2007), or the RM's one-size-fit-all method may not resolve
highly-varied aerosol size impact on AOD calculation. There is another issue of aerosol
specification, which is particular important for organic aerosols as CMAQ 5.1 has 23 SOAs plus
primarily emitted OC (POC). All of organic aerosols were assumed to have the same optical
properties in CRTM and RM, which is obviously an approximate assumption. Liu et al. (2016)
shows that SOA's optical properties could be highly varied depending on the chemical species,
aging time and ambient NOx concentrations.  Both CRTM and RM assume that all aerosols are
externally mixed, which may not best fit the situation in the real world. These uncertainties need
to be addressed in the future verification with observations. In this study, we only proportionally
adjust the aerosol mass concentrations, and did not adjust the aerosol composition, size and
vertical distributions which could have big impact on the AOD calculations. The data
assimilation for initial condition also has its limitation, and its adjustment effect could fade away
with time if there is persistent model bias. So, more frequent adjustment is helpful. Unfortunately
the MODIS AOD assimilation used in this study is applied only once per day in GSI and OI due
to the data availability. More satellite AOD data, such as VIIRS (Visible Infrared Imaging
Radiometer Suite) and GOES-R (Geostationary Operational Environmental Satellite-R Series)
ABT (Advanced Baseline Imager) AODs, should make this assimilation more useful. Another
approach is using more complex and costly four-dimensional (4-D) variational data assimilation
(4D-var) to correct the model's persistent biases, which integrated the data assimilation with the
CTM (Chai, et al., 2016). All these issue should be addressed in the future studies.

## Code Availability

This study includes the forward simulations and data assimilation tools. The meteorological code
of WRF 3.4.1 can be downloaded from
http://www2.mmm.ucar.edu/wrf/users/download/get_source.html. CMAQ 5.1 can be
downloaded from https://www.cmascenter.org/download.cfm. The GSI data assimilation tool can
be downloaded at http://www.dtcenter.org/com-GSI/users.v3.5/downloads/index.php. All other
codes and the modified codes can be provided upon request.



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





Table 1. CMAQ 5.1 aerosol species and their Mapping to GOCART aerosol optical properties in CRTM. (CMAQ species with solid underline are Aitken-mode aerosols, and those with dash underline are coarse-mode aerosols, and the rest are the accumulation-mode aerosols)

| Aerosol Species in CMAQ 5.1 | GOCART aerosols in CRTM |
|---|---|
| ASO4I ANO3I ANH4I ACLI ASO4J ANO3J ANH4J ASO4K ANO3K ANH4K | Sulfate Aerosol |
| AECI  AECJ | Black Carbon Aerosol |
| APOCI APNCOMI APOCJ AOTHRJ AXYL1J AXYL2J AXYL3J ATOL1J ATOL2J ATOL3J ABNZ1J ABNZ2J ABNZ3J AISO1J AISO2J AISO3J ATRP1J ATRP2J ASQTJ AALK1J AALK2J AORGCJ AOLGBJ AOLGAJ  APAH1J APAH2J APAH3J APNCOMJ | Organic Carbon Aerosol |
| AFEJ AALJ ASIJ ACAJ AMGJ AKJ AMNJ ACORS ASOIL | Dust Aerosol |
| ANAJ ACLJ ACLK ASEACAT | Sea Salt Aerosol |





Table 2. Regional statistic of the three simulations (CMAQ_BASE, OI and GSI) for surface PM$_{2.5}$ in July, 2011

| Regions | Simulations | Mean Bias (µg/m$^3$) | Root Mean Square Error (µg/m$^3$) | Correlation Coefficient, R | Index of Agreement |
|---|---|---|---|---|---|
| CONUS | BASE | -2.25 | 10.57 | 0.3 | 0.54 |
| | OI | 0.77 | 11.07 | 0.38 | 0.59 |
| | GSI | -0.73 | 9.42 | 0.44 | 0.64 |
| Northeastern USA | BASE | -3.36 | 11.02 | 0.27 | 0.54 |
| | OI | 0.78 | 11.17 | 0.42 | 0.64 |
| | GSI | -1.18 | 9.70 | 0.44 | 0.66 |
| Pacific Coast | BASE | -2.24 | 8.74 | 0.23 | 0.40 |
| | OI | 0.71 | 9.13 | 0.40 | 0.60 |
| | GSI | -0.80 | 7.86 | 0.44 | 0.61 |
| Southeastern USA | BASE | -1.86 | 11.05 | 0.23 | 0.50 |
| | OI | 0.12 | 9.88 | 0.41 | 0.63 |
| | GSI | -0.58 | 9.30 | 0.41 | 0.63 |
| Rocky Mountain States | BASE | -3.22 | 12.14 | 0.09 | 0.24 |
| | OI | 1.59 | 13.28 | 0.13 | 0.30 |
| | GSI | -1.63 | 11.75 | 0.16 | 0.28 |
| North Central | BASE | -1.55 | 12.29 | 0.22 | 0.49 |
| | OI | 0.96 | 13.83 | 0.28 | 0.51 |
| | GSI | -0.26 | 11.12 | 0.35 | 0.59 |
| South Central | BASE | -1.1 | 9.49 | 0.09 | 0.4 |
| | OI | 0.38 | 8.07 | 0.27 | 0.52 |
| | GSI | -0.1 | 7.58 | 0.27 | 0.54 |




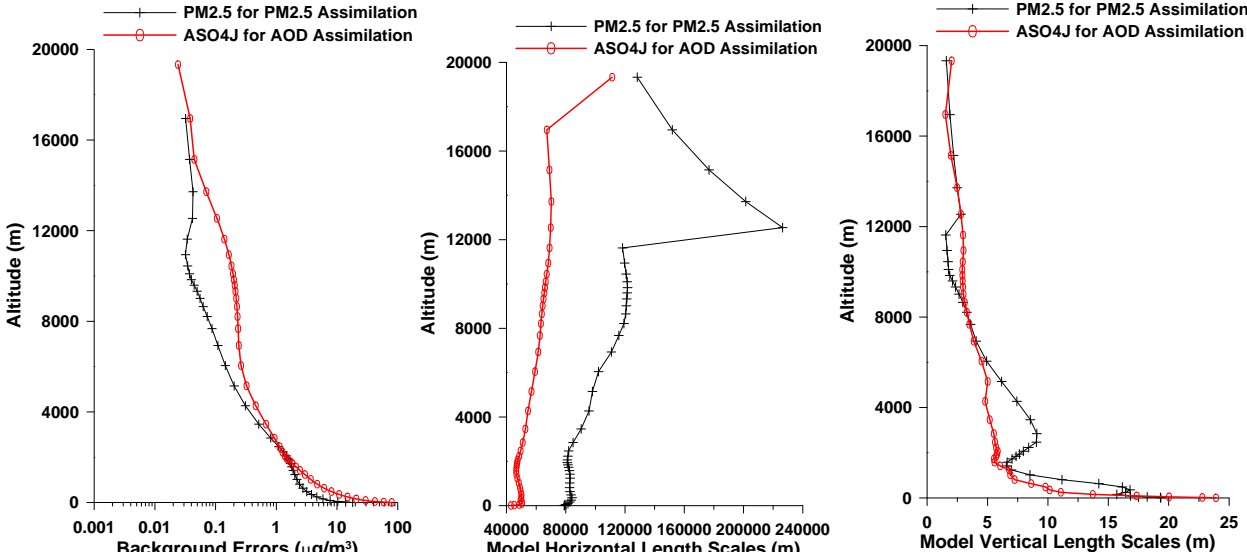

Figure 1. GSI's background errors and length scales used in this study (ASO4J is the CMAQ's accumulation-mode sulfate aerosol)



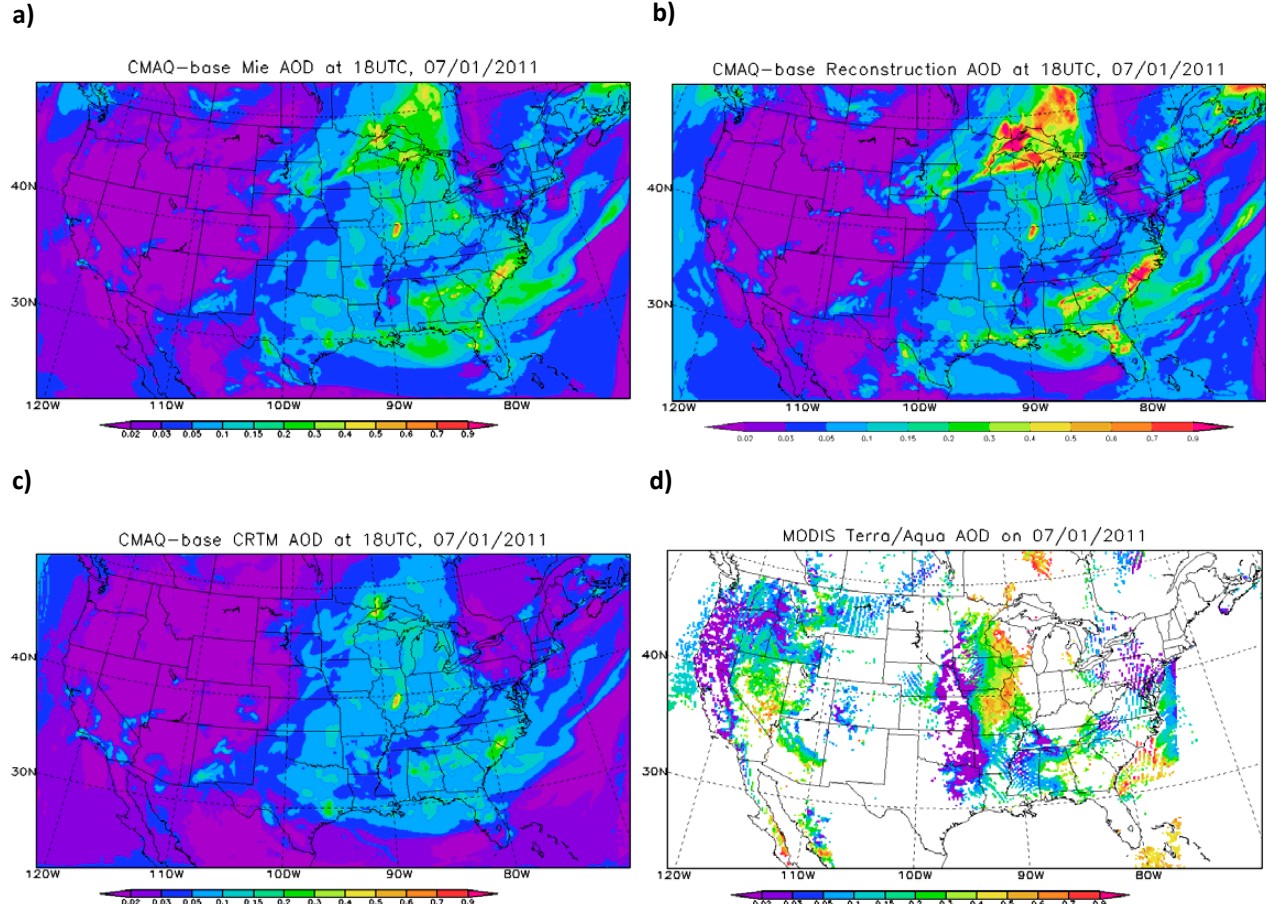

Figure 2, AOD calculations using Mie-method (a), reconstruction method (b), and CRTM from the CMAQ's base case (before assimilations) compared to MODIS AOD on 07/01/2011.





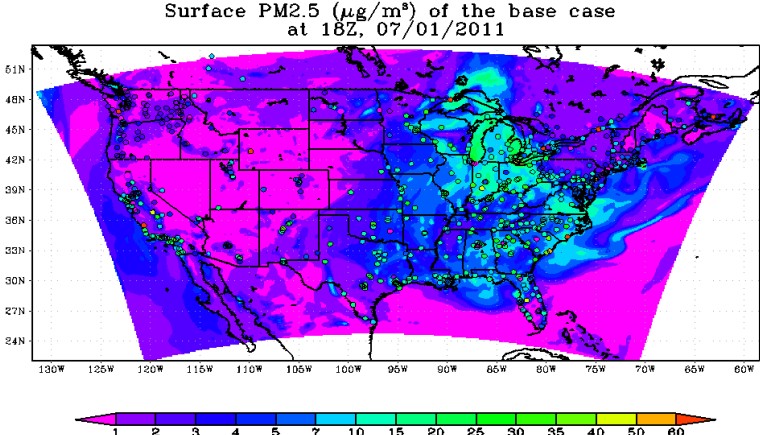

Figure 3. Predicted surface PM$_{2.5}$ from CMAQ base run compared to surface measurement (close circle) at 18 UTC, 07/01/2011.





Figure 4. surface accumulation-mode sulfate (ASO4J) changes after GSI (left) and OI (right) assimilations with surface PM$_{2.5}$ and AOD.





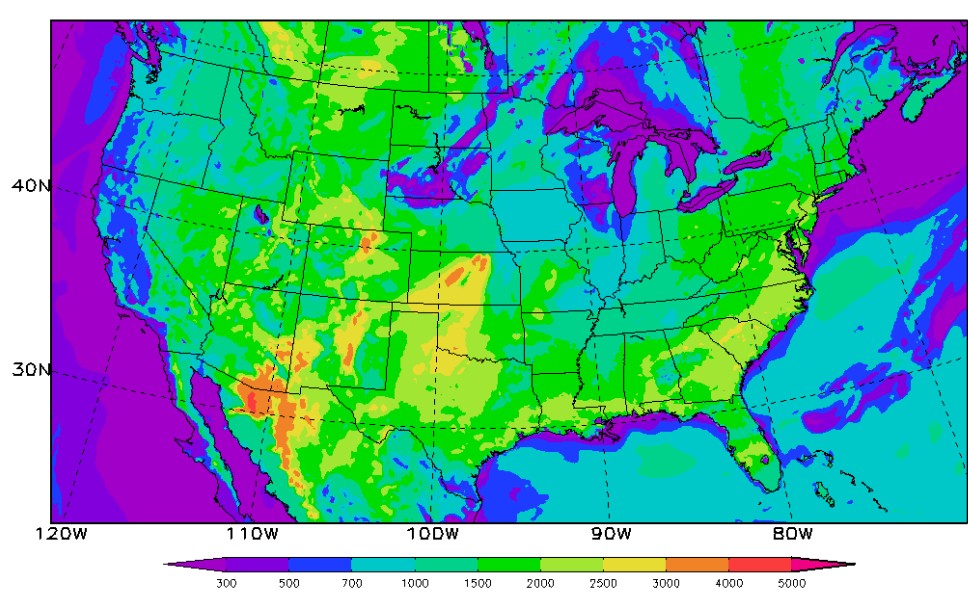

Figure 5. Model-Predicted PBL height (m) at 18UTC, 07/01/2011



Figure 6, same as the figure 4 but for the model's 2km layer.



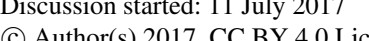


Figure 7, same as the Figure 4 but for AOD change. Left plots show the CRTM
AOD change due to GSI adjustments and right plots are RM AOD change for OI.



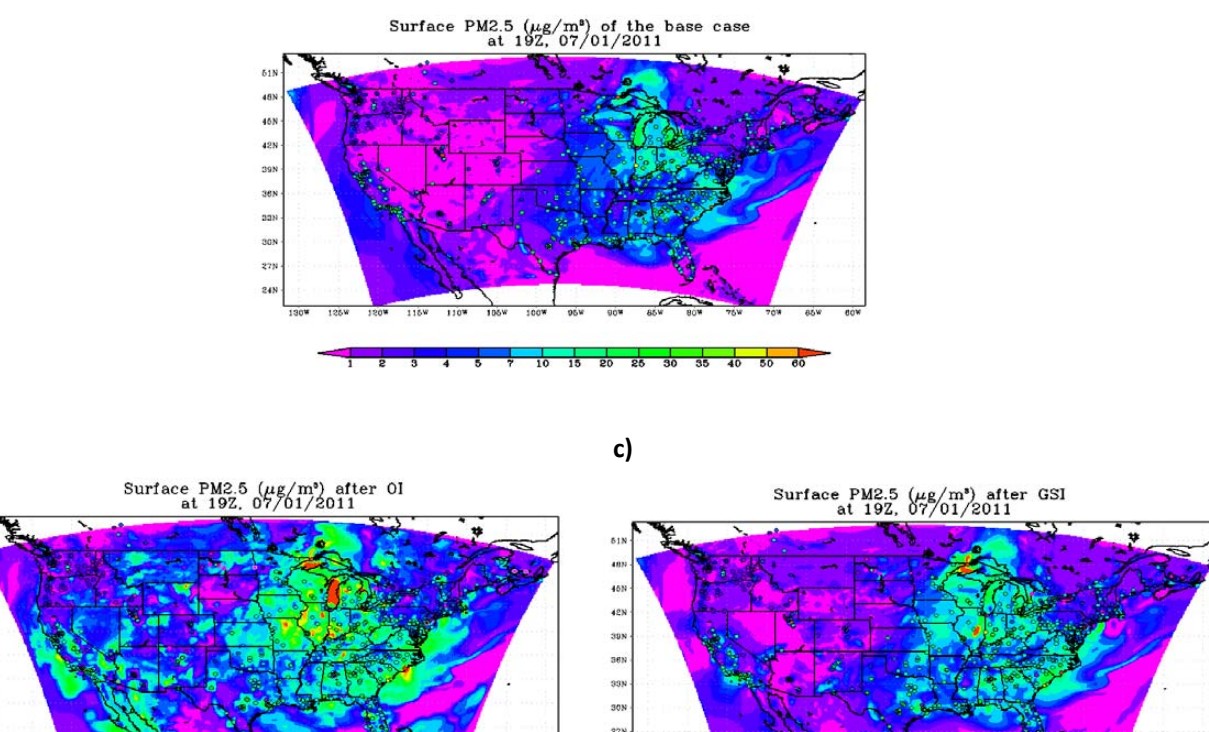

Figure 8, Predicted surface PM$_{2.5}$ from the base case (a), OI (b) and GSI (c) runs compared to surface observation at 19UTC, 07/01/2011.

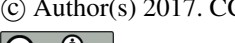



Figure 9, Time-series comparisons of CMAQ surface PM$_{2.5}$ (base, OI, and GSI) versus the observation over the CONUS domain for their mean value (a), correlation coefficient (b) and root mean squared error (RMSE) (c)