# Peer review of "A Case Study of Aerosol Data Assimilation with the Community Multi-Scale"

_Geoscientific Model Development, 2017_

## Referee Comment (RC1) · Anonymous Referee #1 · 12 Jul 2017

The authors discusses the differences between two assimilations systems based on GSI and OI using the same model and the same set of observations. However, the comparison maybe unfair if the error statistics used in the two systems are different by an appreciable margin. It is not entirely clear from the document if the same error statistics is used, or how are they different. I would thus suggest that section 2.1 pay lot more attention to this issue. A table could be useful. Also Figure 1 only displays background error characteristics for the GSI system. Could the OI error variances and lengths scales be displayed in reference to a Gaussian model for both. Showing a one obs. impact from each analysis system would be also informative.

---

## Short Comment (SC1) · 12 Jul 2017

ïČŸ Thank you for your prompt comments. In this study, the observations and observation errors (constant 0.1) used in these two assimilation tools are same. However, their model errors are not the same, which is partly due to their different engineering structures. As we mentioned that GSI is 3D-var system, and its error covariance and length scales have vertical variation as shown in Figure 1. The OI assimilation is made in horizontally in 11×11 grid box (12km resolution each grid), so its horizontal length scale is approximately 11/2×12 =66km. Vertically, OI applies its adjustment ratio up to PBL top for surface PM2.5 assimilation, or whole column for AOD assimilation over

each grid location. OI's model errors for surface PM2.5 assimilation have diurnal variation and are varied from location to location based on the raw run's statistics, which is carried out in this study (Figure 2 of Tang et al., 2015). The GSI's model errors have no temporal variation or horizontal changes, but have vertical variations. Their biggest behaviors difference can be seen from the Figures 4a,b: the OI assimilation mainly affect local or nearby grids while GSI 3D-var increment expands more broadly.

[Figure]

[Figure]

Model Relative Uncertainties use in OI's surface PM2.5 assimilation at 17 UTC

Figure 2 of Tang, Y., T. Chai, L. Pan, P. Lee, Daniel Tong, H. Kim, and W. Chen, Using Optimal
Interpolation to Assimilate Surface Measurements and Satellite AOD for Ozone and PM2.5: A Case Study
for July 2011. Journal of the Air & Waste Management Association,
DOI:10.1080/10962247.2015.1062439, 2015

---

## Referee Comment (RC2) · Anonymous Referee #2 · 16 Aug 2017

General recommendation

The general topic of the paper is a comparison of the performance of two different data assimilation methods when using the same air quality model and AOD/PM2.5 observations. Although both data assimilation methods have been used with air quality models before, the models themselves have been different (CMAQ and GOCART) and so a direct comparison on their relative performance has not been possible. This paper discusses the different set up of each of the data assimilation schemes and looks at the results of a single assimilation cycle and then the more general performance over the month of July 2011. The differences between the results of the two data assimilation

schemes are linked to the difference in the set-up of the methods, in particular for the single assimilation cycle.

Overall the paper contains interesting and relevant information and results. Although the data assimilation methods are not new concepts or ideas, to see them compared in a more consistent manner provides useful information for anyone looking to establish an air quality model that utilises observations. Particularly in regards to the different scenarios where the schemes show stronger or weaker performance. My main criticism of the paper is that the descriptions of how the actual experiments have been set-up are not sufficient for someone without an in-depth knowledge of the models and/or the data assimilation code to understand exactly what has been done. I detail my specific queries in the major comments below.

Major comments

1. Section 2.1 – page 3, lines 13 to 16 and 23 to 24. There are two queries I have here about the OI set up that I believe are related. (i) The first is that I'm not clear how the background uncertainties have been formulated. I've also looked at the referenced paper but still have questions and so I think that a brief but slightly more comprehensive description would be useful here. As far as I can tell from the referenced paper, the background uncertainties have been made using a free running model and comparing to the observations. If I look at Figure 2 from that paper then this only gives values at observation locations. Is this correct? Generally B is formulated to give uncertainty values across the whole of the model domain not just at point locations, so why is this not done in this case? You also mention the diurnal variations – does this mean you use a different B matrix for the different assimilation cycles? This also only addresses the diagonal variances or uncertainties of the B matrix and not the cross correlations and this is my second question. (ii) You mention that the OI adjustment is made in each 11x11 grid horizontally and it's effect expands up to the PBL. This would normally be done through the B matrix, is this the case here? If so is it a cut-off Gaussian that has been used for the horizontal correlations, or a uniform distribution? Similarly for the

vertical correlations – you state that the surface PM2.5 OI increment is applied from the surface to the height of the PBL. Is this done after an increment is calculated by OI or through the formulation of the B matrix? Is it a constant adjustment at each model level or does it taper as you reach the PBL? If so how does it taper? All this is relevant because you attribute the differences in the schemes largely to this formulation of the B matrix.

2. Section 2.1 – page 4, lines 1 to 4. Again this question addresses the B matrix but now for the GSI scheme. The referenced paper was informative and from this I understand that the horizontal and vertical correlations that make up the B matrix are Gaussian (and again it might be worth mentioning that here) and I assume the GSI package has been used to create these? Although it talks about optimisation in the referenced paper, I couldn't find anything to suggest these length-scales should be the same across the domain. Are the background errors and length scales the same across the domain in this case? It is unclear from the description given whether these plots are a single point representation or not. I also wasn't totally clear what is being used as the control variable for the AOD. Is it just ASO4J? You also mention that other aerosol species have proportional model or background errors. Were these just computed for interest or do they also feed in to the modelled AOD?

3. Section 2.2 – page 5, lines 17-40. The final issue that I really struggled to understand was how the CMAQ forecast model and the GSI data assimilation method actually interact. Perhaps a schematic attached to this section would be more informative, as it's clearly quite a complex process. Are the species given in the forecast by CMAQ, translated to the variables required by the CRTM's GOCART aerosol as stated in Table 1, then run through the forward model calculation to give AOD, assimilated by GSI and an increment returned for each of the GOCART species? How is this increment divided back in to the CMAQ variables? Once this is done I assume that CMAQ is used to forecast the species on to the next assimilation time? Is the B matrix you describe also a plug in to the GSI? Is the resolution the same for both models? If a

figure could be created that shows this process it would add a lot to the understanding of this section

4. Section 3.1 – page 7. How do you distinguish the effect of assimilating AOD from the effect of the PM2.5 measurements? Are three different experiments run (one with just PM2.5, one with AOD and one with both)? Given that the GSI method solves a cost function, I'm not clear how you distinguish the effects from two different observation types in one experiment. For the OI, as far as I understand an increment is given by the AOD and PM2.5 observations separately, although this needs to be described/clarified in further detail (see my question 1 above), and so I can potentially envisage how this could be done with one experiment. However, this needs to be made clear.

Minor comments

1. Abstract – page 1, line 26. I think perhaps 'background error uncertainties and the horizontal/vertical length-scales of the covariances' would be a more accurate description.

2. Figure 1. Are these errors and length-scales static quantities used across the domain or for one specific point.

3. Figure 4 is in the wrong order.

4. Page 9, line 14 - 'assimilation for the elevated layers blow PBL.'. Should be below.

5. Page 9, line 10 - 'steeper' not 'steepen'

---

## Referee Comment (RC3) · Anonymous Referee #1 · 22 Aug 2017

This paper is really about the Implementation of aerosol assimilation with GSI using the CMAQv5.1 model. As a way of comparison it compares the results with a previous aerosol assimilation system developed by Tang et al. (2015) which provides material for an interesting discussion, but it is in no way a formal or even valid comparison between a 3DVar and OI as the title suggest. The two assimilation systems differ: 1- by their analysis method, 2- by the method to obtain analysis increments (error statistics for GSI, versus prescribed influence functions for Tang et al. (2015)), and 3 - for matching model representation with radiative transfer model and observables. They are thus too many aspects at stakes to be able to draw any informed conclusion about the merits of 3D Var against OI or any other analysis scheme. Furthermore, the system used

in Tang et al. (2015) is not an OI but is rather a (data) nudging method. OI is based on covariance functions. The influence functions used by Tang et al. (2015) are boxes of 11x11 grid points horizontally and with PBL height in the vertical (for PM2.5 observation assimilation), and cannot be derived from any covariance functions, as they are not positive definite. I thus do not recommend the publication of this manuscript as a comparison between two analysis systems and would strictly avoid presenting conclusions in that sense, but rather would focus the paper about the implementation of aerosols assimilation with GSI using CMAQ. However, instead of assessing an assimilation system by comparing with observations, as it is normally done or by examining its impact on forecast, in the case of aerosols, the aerosols observations gives in fact too little information to draw upon. The interesting concept used in this paper is to compare instead with a simple assimilation system in order to draw conclusions. I realize that this change of focus amounts to a significant rewriting of the manuscript, but it is in my opinion the only fair option that would avoid misleading conclusions. The introduction should also adopt a terminology that is more in line with current practice to avoid expressions like "Another method is indirect guessing" (line 14, p2).

---

## Referee Comment (RC4) · Anonymous Referee #3 · 23 Aug 2017

Aerosol data assimilation in Weather-Chemistry coupled models has been increasingly practiced to improve air quality forecast with satellite aerosol observations. This article compares the 3D-Var and OI data assimilation methods applied to PM2.5 and/or AOD observations in the CMAQ modeling system. This is an interesting work, but the quality of the paper still has room to be be improved.

Major comments:

First, the purpose of data assimilation is to improve forecast. However, this paper presents mainly the impact of data assimilation on the simulation at the assimilation time window. It does show impact on prediction, but it was a only one-hour forecast

from 18 to 19 UTC on 07/01/2011 (Figure 8). Additionally, it is unclear to the reviewer if the time-series plot (Figure 9) showed forecast results or adjusted results from assimilating data on the same time window.

Second, the data used for verification is not clearly described. Are they the same PM data used in the assimilation? Please note that the data used for assimilation should not be used in the verification of data assimilation process. Otherwise, the verification is cheating by checking self-consistency.

Specific comments:

1.Treatments on observation and prior errors need a better justification. Error characterization on observation and background are essential to data assimilation. Observation error was simply assumed as constant (0.1) for both PM2.5 and AOD. This needs to be justified with error covariances.

2.The data assimilation system should be described in more detail. In particular, the OI system needs a detailed description, so that readers does not need to go to Tang et al 2015 for essential information about the method. Also, it is not clear how CMAQ and GSI 3D-Var get coupled.

3.A brief description and references of the PM2.5 data are needed. How many number of sites are used? Were all sites used for both data assimilation and model evaluation?

4.A brief description of MODIS AOD is also necessary with references. It shall include which MODIS product is used (e.g., level 2 or level 3, which collection, on which wavelength) and why 18Z data assimilation is applied, etc.

5.Page 6, line 34: Do those mass scaling factor vary with time and location? If they are constants, please provide here.

6.Figure 2: It is not clear to the reader which wavelength is for these AODs. I cloud only confirm that the Reconstruction AOD (panel b) is at 550 nm.

[Figure]

7.Figure 4: Arrangement of panels in Figure 4 is confusing and please make them clear and consistent. The figure caption says GSI is on the left and OI on the right, but OI is found in the middle row for both columns and GSI is found at the bottom row for both columns too. According to the text, I think those panels should be arranged like Figure 6: (a) GSI_PM, (b) OI_PM, (c) GSI_AOD, (d) OI_AOD, (e) GSI_All, (f) OI_All.

8.Figure 7: Again, please specify the used spectral wavelength for AODs.

9.Evaluation of model predictions needs further quantitative statistically analysis. For instance, bias, RMSE and R^2 for predictions of PM2.5 against observation should be reported.

10. Font size in almost every plot needs to be increased, especially for color-bar axes.

---

## Author Comment (AC1) · 2 Oct 2017

We agree that this manuscript should not be treated as the general comparison of 3d-Var and OI data assimilations. Instead, it is just a regional case study with these specified methods. We changed the title accordingly (StatesA case study of Aerosol data assimilation with the Community Multi-scale Air Quality Model over the Contiguous United States using 3D-Var and optimal interpolation methods) as well as theabstract and corresponding conclusion part. We understand the reviewer's concern on the OI method. It indeed has some nudging flavor in the system, such as limiting the PM2.5 assimilation below the PBL and only applying the AOD assimilation above it

[Figure]

when both AOD and PM2.5 observations are available at one grid cell. However, we believe it is essentially still an OI scheme, based on covariance functions (see Chai et al. 2017, for detail,http://onlinelibrary.wiley.com/doi/10.1002/2016JD026295/full). The boxes of 11x11 grid points horizontally and with PBL height in the vertical direction (for PM2.5 observation assimilation) are the localization applied for the efficiency of the computation.

---

## Author Comment (AC2) · 2 Oct 2017

Thank you for your detailed comments, which highlighted some important technique details. We changed this manuscript and make it easy to understand.

*Section 2.1 – page 3, lines 13 to 16 and 23 to 24. There are two queries I have here about the OI set up that I believe are related. (i) The first is that I'm not clear how the background uncertainties have been formulated. I've also looked at the referenced paper but still have questions and so I think that a brief but slightly more comprehensive description would be useful here. As far as I can tell from the referenced paper, the background uncertainties have been made using a free running model and comparing*

*to the observations. If I look at Figure 2 from that paper then this only gives values at observation locations. Is this correct? Generally B is formulated to give uncertainty values across the whole of the model domain not just at point locations, so why is this not done in this case?*

It is true that the background error covariance matrix B needs to be formulated to give uncertainty values across the whole model domain. The dynamic uncertainty used in Tang et al. (2015) is applied to the locations in which surface PM2.5 monitoring stations exist, while all other areas use 80

*You also mention the diurnal variations – does this mean you use a different B matrix for the different assimilation cycles? This also only addresses the diagonal variances or uncertainties of the B matrix and not the cross correlations and this is my second question.*

Yes, the variances of the B matrix will change for different assimilation cycles over the monitoring stations. However, the cross correlations are only modeled as a function of separation distance, which has no diurnal variation (see equation 3 of Chai et al. 2017, http://onlinelibrary.wiley.com/doi/10.1002/2016JD026295/full ). The corresponding clarification was added in the manuscript.

*(ii) You mention that the OI adjustment is made in each 11x11 grid horizontally and it's effect expands up to the PBL. This would normally be done through the B matrix, is this the case here? If so is it a cut-off Gaussian that has been used for the horizontal correlations, or a uniform distribution? Similarly for the vertical correlations – you state that the surface PM2.5 OI increment is applied from the surface to the height of the PBL. Is this done after an increment is calculated by OI or through the formulation of the B matrix? Is it a constant adjustment at each model level or does it taper as you reach the PBL? If so how does it taper? All this is relevant because you attribute the differences in the schemes largely to this formulation of the B matrix.*

It is a cut-off Gaussian-like B matrix in the horizontal directions, but uniform distribution

for the adjusting ratio in the vertical direction up until the PBL (or whole column for AOD). At each grid point, an increment is calculated using OI formulation and the adjustment ratio is applied to all levels below the PBL (or whole column for AOD). No tapering is applied here.

*Section 2.1 – page 4, lines 1 to 4. Again this question addresses the B matrix but now for the GSI scheme. The referenced paper was informative and from this I understand that the horizontal and vertical correlations that make up the B matrix are Gaussian (and again it might be worth mentioning that here) and I assume the GSI package has been used to create these? Although it talks about optimisation in the referenced paper, I couldn't find anything to suggest these length-scales should be the same across the domain. Are the background errors and length scales the same across the domain in this case? It is unclear from the description given whether these plots are a single point representation or not. I also wasn't totally clear what is being used as the control variable for the AOD. Is it just ASO4J? You also mention that other aerosol species have proportional model or background errors. Were these just computed for interest or do they also feed in to the modelled AOD?*

Yes, B matrix is Gaussian here, which is generated from a GSI utility, called GEN_BE. You can find more detail about the tool (Descombes, et al, 2015, and http://www.dtcenter.org/com-GSI/users.v3.5/docs/presentations/2014_tutorial/Auligne_GSI_Tutorial2014.pdf).
Yes, the background errors and length scales used in this study are same across the domain for each control variable, and only have vertical variations. In AOD assimilation, the 54 control variables are listed in Table 1 (CMAQ 5.1 aerosol species, left column), and ASO4J is just one of them. We changed the "proportional" to "similar" to avoid from misunderstanding. Each aerosol species in AOD assimilation actually has its own background error and length scales. They are similar in term of B's vertical variation in the same aerosol size mode, but their standard deviations have magnitude difference due to their different atmospheric abundance. We added the clarification in

manuscript.

*Section 2.2 – page 5, lines 17-40. The final issue that I really struggled to understand was how the CMAQ forecast model and the GSI data assimilation method actually interact. Perhaps a schematic attached to this section would be more informative, as it's clearly quite a complex process. Are the species given in the forecast by CMAQ, translated to the variables required by the CRTM's GOCART aerosol as stated in Table 1, then run through the forward model calculation to give AOD, assimilated by GSI and an increment returned for each of the GOCART species? How is this increment divided back in to the CMAQ variables? Once this is done I assume that CMAQ is used to forecast the species on to the next assimilation time? Is the B matrix you describe also a plug in to the GSI? Is the resolution the same for both models? If a figure could be created that shows this process it would add a lot to the understanding of this section* We add the schematic diagram, changed the literature and make it easy to understand. The GSI AOD assimilation actually uses the 54 CMAQ species, not the GOCART species. The concentrations of 54 species are feed into GSI and also gotten their each increment out from GSI. We only use GOCART's lookup table in CRTM to convert each CMAQ species' mass concentration to each aerosol optical properties etc which are needed for the AOD assimilation. Table 1 is used for the AOD conversion only. So, there is no species concentration re-distribution or model resolution issue, as all of them are native.

*Section 3.1 – page 7. How do you distinguish the effect of assimilating AOD from the effect of the PM2.5 measurements? Are three different experiments run (one with just PM2.5, one with AOD and one with both)? Given that the GSI method solves a cost function, I'm not clear how you distinguish the effects from two different observation types in one experiment. For the OI, as far as I understand an increment is given by the AOD and PM2.5 observations separately, although this needs to be described/clarified in further detail (see my question 1 above), and so I can potentially envisage how this could be done with one experiment. However, this needs to be made clear.*

Yes, we had three experiments to showing the effect, one with just PM2.5, one with AOD and one with both. You are right, and it is not easy to treat two different observation types in existing GSI in one step. For 18Z GSI assimilation, the assimilation was actually made in two steps: first is the GSI AOD assimilation, and surface PM2.5 assimilation was made upon the AOD assimilation adjustment. We added this content in the manuscript.

*Abstract – page 1, line 26. I think perhaps 'background error uncertainties and the horizontal/vertical length-scales of the covariances' would be a more accurate description.* Changed. Thank you

*Figure 1. Are these errors and length-scales static quantities used across the domain or for one specific point.*

They are used across the domain. We added clarification.

*3. Figure 4 is in the wrong order. 4. Page 9, line 14 - 'assimilation for the elevated layers blow PBL.'. Should be below. 5. Page 9, line 10 - 'steeper' not 'steepen'*

Changed. Thank you

Thank you again for your comments

---

## Author Comment (AC3) · 2 Oct 2017

Thank you for your comments. We revised this manuscript accordingly. Here are the answers to your specified questions.

*First, the purpose of data assimilation is to improve forecast. However, this paper presents mainly the impact of data assimilation on the simulation at the assimilation time window. It does show impact on prediction, but it was a only one-hour forecast from 18 to 19 UTC on 07/01/2011 (Figure 8). Additionally, it is unclear to the reviewer if the time-series plot (Figure 9) showed forecast results or adjusted results from assimilating data on the same time window. Second, the data used for verification is not*

*clearly described. Are they the same PM data used in the assimilation? Please note that the data used for assimilation should not be used in the verification of data assimilation process. Otherwise, the verification is cheating by checking self-consistency.*

The data assimilation methods used in study are mainly about adjusting initial condition. So, its immediate impact is the change of initial condition, which was discussed in Figures 4-7. Its impact on next-hour forecast was discussed in Figure 8. All thereafter impacts were shown in figure 9 and table 2, for one-month performance in 4-cycle per day. The data used in verification are not those used in data assimilation. For 18UTC data assimilation and forecast, the verification starts from 19UTC to 00UTC, and 00UTC cycle's verification is done from 01UTC to 06UTC. We added this clarification in the manuscript.

*1.Treatments on observation and prior errors need a better justification. Error characterization on observation and background are essential to data assimilation. Observation error was simply assumed as constant (0.1) for both PM2.5 and AOD. This needs to be justified with error covariances*

Thank you for your comments. We added some justification about the observation error.

*2. The data assimilation system should be described in more detail. In particular, the OI system needs a detailed description, so that readers does not need to go to Tang et al 2015 for essential information about the method. Also, it is not clear how CMAQ and GSI 3D-Var get coupled*

Thank you for your suggestion. We added more detailed about the OI and how CMAQ and GSI 3D-Var get coupled.

*3. A brief description and references of the PM2.5 data are needed. How many number of sites are used? Were all sites used for both data assimilation and model evaluation.*

Good suggestion. We added this information

*4. A brief description of MODIS AOD is also necessary with references. It shall include which MODIS product is used (e.g., level 2 or level 3, which collection, on which wavelength) and why 18Z data assimilation is applied, etc.*

We added the information about MODIS AOD data

*5. Page 6, line 34: Do those mass scaling factor vary with time and location? If they are constants, please provide here.*

The mass allocation factors of PM25AT, PM25AC, and PM25CO are varied with time and location, which depends on the aerosol size distributions in the three modes. We added the clarification

*6. Figure 2: It is not clear to the reader which wavelength is for these AODs. I could only confirm that the Reconstruction AOD (panel b) is at 550 nm*

All of the AOD are in 550nm. We added the clarification

*Figure 4: Arrangement of panels in Figure 4 is confusing and please make them clear and consistent. The figure caption says GSI is on the left and OI on the right, but OI is found in the middle row for both columns and GSI is found at the bottom row for both columns too. According to the text, I think those panels should be arranged like Figure 6: (a) GSI_PM, (b) OI_PM, (c) GSI_AOD, (d) OI_AOD, (e) GSI_All, (f) OI_All*

You are right. We re-arranged the plot panels.

*Figure 7: Again, please specify the used spectral wavelength for AODs.*

We added

*Evaluation of model predictions needs further quantitative statistically analysis. For instance, bias, RMSE and RËȨ2 for predictions of PM2.5 against observation should be reported.*

The statistics of bias, RMSE and R are listed in table 2

*Font size in almost every plot needs to be increased, especially for color-bar axes*

We increased the fonts of those figures.

Thank you again for your comments